# Plasma and Urine Circulating Tumor DNA Methylation Profiles for Non-Invasive Pancreatic Ductal Adenocarcinoma Detection: Significant Findings in Plasma Only

**DOI:** 10.3390/ijms26114972

**Published:** 2025-05-22

**Authors:** Tomoaki Ito, Takumi Iwasawa, Shunsuke Sakuraba, Kenichiro Tanaka

**Affiliations:** 1Department of Surgery, Juntendo University Shizuoka Hospital, Juntendo University School of Medicine, Shizuoka 410-2295, Japan; sswatana@juntendo.ac.jp (S.S.); kn-tanaka@juntendo.ac.jp (K.T.); 2Shizuoka Medical Research Center for Disaster, Juntendo University, Shizuoka 410-2295, Japan; iwasawa@toyo.jp; 3Institute of Life Innovation Studies, Toyo University, Tokyo 115-8650, Japan

**Keywords:** pancreatic ductal adenocarcinoma, circulating tumor DNA (ctDNA), DNA methylation, liquid biopsy, whole-genome bisulfite sequencing (WGBS)

## Abstract

Pancreatic ductal adenocarcinoma (PDAC) is a lethal malignancy with limited treatment options, and early detection remains challenging due to the lack of reliable non-invasive biomarkers. This study investigated plasma and urine circulating tumor deoxyribonucleic acid (ctDNA) methylation profiles as potential biomarkers for PDAC detection. A total of 35 patients with PDAC and 10 non-cancerous controls were enrolled, and whole-genome bisulfite sequencing was performed on ctDNA extracted from both plasma and urine samples. Plasma ctDNA methylation profiles effectively distinguished cancerous from non-cancerous samples, particularly through differential methylation in intergenic regions. Hierarchical clustering further enabled the accurate grouping of patients with PDAC. However, urine ctDNA did not show significant methylation differences. These findings suggest that plasma ctDNA methylation holds promise as a non-invasive biomarker for PDAC detection, whereas urine ctDNA appears less informative. Future research should validate these findings in larger cohorts and investigate machine learning approaches to improve diagnostic performance, ultimately facilitating earlier detection and improved patient outcomes.

## 1. Introduction

Pancreatic ductal adenocarcinoma (PDAC) remains one of the most aggressive malignancies, characterized by a dismal prognosis and limited treatment options. Current diagnostic methods for PDAC predominantly rely on invasive procedures such as endoscopic ultrasound-guided fine-needle aspiration (EUS-FNA) and endoscopic retrograde cholangiopancreatography (ERCP). Although these techniques are instrumental, false-negative results and potential complications are frequently associated with their use, emphasizing the pressing need for more reliable and less invasive diagnostic approaches [1].

The quest for effective blood-based biomarkers has led to the clinical adoption of carbohydrate antigen 19-9 (CA19-9). However, the ability of CA19-9 to detect early-stage PDAC remains limited due to its elevation in non-malignant conditions and pancreaticobiliary diseases, underscoring the necessity for more specific markers [2]. At the genetic level, PDAC is distinguished by frequent oncogenic mutations in *KRAS* and loss-of-function mutations in tumor suppressors such as *TP53*, *CDNK2A*, *DPC4/SMAD4*, and *BRCA2* [3]. These genetic alterations have fueled research into circulating tumor deoxyribonucleic acid (ctDNA) as a potential non-invasive biomarker.

ctDNA, a subset of cell-free DNA (cfDNA) found in blood, carries cancer-specific genetic alterations [4]. Advancements in detection technologies have positioned ctDNA as a promising biomarker for cancer diagnosis [5]. ctDNA analysis shows potential in non-invasive cancer detection, prognosis, monitoring therapy responses, and tracking tumor evolution [4,6]. Various methods, including polymerase chain reaction (PCR)-based techniques and next-generation sequencing (NGS), have been developed for ctDNA analysis [7]. However, detection rates vary depending on the disease stage and methodologies employed, ranging from 30–50% in resectable tumors to 50–100% in advanced cases [8,9].

Although plasma ctDNA has demonstrated potential as a non-invasive biomarker, urine-based liquid biopsy offers advantages such as ease of collection and suitability for more frequent sampling, making this approach attractive for cancer detection, monitoring, and personalized medicine [10,11]. Unlike blood-based biopsies, urine sampling enables more frequent and less invasive testing [12]. Urine contains various biomarkers, including ctDNA, exosomes, and ribonucleic acid (RNA) molecules, that can be analyzed [13,14]. Studies have demonstrated that urine-based liquid biopsy is applicable to both urological and non-urological cancers, such as colorectal cancer, hepatocellular carcinoma, non-small cell lung cancer, stomach cancer, and breast cancer, with sensitivities comparable to blood-based methods [4,15]. In PDAC specifically, Terasawa et al. reported the potential of liquid biopsy using both plasma and urine for diagnosis, detecting *KRAS* mutations in both sample types with a detection rate of approximately 48% [16]. However, the clinical utility of urine-based liquid biopsy in PDAC remains unclear.

To improve diagnostic accuracy in liquid biopsy, this study focused on DNA methylation. Epigenetic alterations, particularly abnormal DNA methylation in promoter regions of tumor suppressor genes, play a critical role in cancer progression [17]. Consequently, interest in methylation-based diagnostic approaches continues to grow. Most studies have concentrated on locus-specific or CpG-rich genomic regions, using technologies like methylated DNA immunoprecipitation sequencing (MeDIP-seq) and reduced representation bisulfite sequencing (RRBS) [18,19]. However, a comprehensive understanding of genome-wide DNA methylation in cfDNA at single-base resolution, comparing malignant and non-malignant conditions, remains incomplete. Whole-genome bisulfite sequencing (WGBS) enables detailed assessment of methylation across the entire genome at single-base resolution, offering a promising strategy to develop novel diagnostic and prognostic tools for PDAC that could transform early detection and treatment [20].

Given the limitations of current biomarkers such as CA19-9 and the growing evidence supporting the diagnostic utility of ctDNA methylation, this study hypothesized that methylated DNA could be effectively detected in the plasma and urine of patients with PDAC using liquid biopsy. By employing WGBS, the study aimed to identify methylation signatures capable of distinguishing PDAC from non-cancerous conditions and to evaluate the relative effectiveness of plasma and urine for early detection.

## 2. Results

### 2.1. Methylation Profiles of Plasma ctDNA Differ Between Patients with PDAC and Non-Cancer Controls

WGBS using NGS was performed on ctDNA extracted from plasma samples of patients with PDAC and non-cancer controls (Figure 1A).

A comparison of 10 non-cancer cases and 35 PDAC cases using urinary ctDNA revealed subtle differences in ctDNA methylation profile distributions between the two groups. These findings suggest the potential presence of methylation sites with diagnostic relevance (Figure 1B,C). Fisher’s exact test identified 8 CpG sites with significant hypermethylation and 116 CpG sites with significant hypomethylation in PDAC cases. All CpG sites were located in intergenic regions (Appendix A).

### 2.2. Differential Methylation Profiles of Plasma ctDNA Do Not Correlate with Existing Tumor Markers

Hierarchical clustering analysis revealed a distinct cluster consisting exclusively of PDAC cases (cluster A). A comparison of tumor marker levels—CEA, CA19-9, DUPAN-2, and Span-1—between cluster A and a separate cluster (cluster B) showed no significant differences. These results showed that existing tumor markers did not differ between cluster A and cluster B, suggesting that the methylation profile of plasma ctDNA could be a new candidate biomarker for PDAC (Figure 2). Additionally, an exploratory analysis of clinical variables such as age, sex, tumor size, and stage revealed no apparent differences between cluster A and cluster B.

### 2.3. Methylation Profiles of Urinary ctDNA Do Not Differ Between Patients with PDAC and Non-Cancer Controls

Analysis of urinary ctDNA methylation using NGS revealed limited differential methylation between the patients with PDAC and the non-cancer controls. Among the 35 PDAC cases and 10 non-cancer controls, 7 CpG sites showed significantly increased methylation, and 1 site displayed significantly decreased methylation (Appendix A). All differentially methylated sites were located in intergenic regions. However, clustering analysis failed to distinguish PDAC from control cases based on urinary ctDNA methylation profiles (Figure 3A,B).

The methylation patterns observed in urine were completely distinct from those in plasma, indicating a clear differentiation between the two sample types.

## 3. Discussion

### 3.1. Methylation Profiles of Plasma and Urinary ctDNA

DNA methylation has emerged as one of the most well-studied cancer-associated epigenetic changes suitable for use as a biomarker of cancer [21]. Cancer cells exhibit both global hypomethylation, which contributes to genomic instability, and focal hypermethylation, particularly at gene promoters, resulting in the silencing of tumor suppressor genes [21,22]. These methylation changes arise early during carcinogenesis, making them promising candidates for early cancer detection [21,22]. The current study demonstrated that plasma ctDNA methylation profiles serve as potential biomarkers for PDAC detection, with significant differences observed between patients with PDAC and non-cancerous controls. Although high specificity was observed in plasma ctDNA methylation, sensitivity remained limited, as only a subset of patients with PDAC exhibited detectable changes. This finding may reflect biological heterogeneity or low ctDNA shedding in some cases, suggesting the occurrence of false-negative results. Indeed, only 10 out of 35 patients with PDAC were classified as positive based on methylation profiling, indicating that the current approach identifies only a subset of true cases. Despite the absence of false positives, the high number of false negatives highlights the sensitivity limitations. These findings underscore the need for incorporating additional biomarkers or adopting multi-omic strategies to improve detection rates.

Furthermore, methylated DNA retains stability under various storage conditions and remains analyzable even when highly fragmented, making it a viable target in urine samples. Several studies have successfully detected methylation signals in urine from patients with non-urological cancers, such as cervical [23], colorectal [24], endometrial [25], liver [26], lung [27], and ovarian [28] cancers, using PCR-based methods. However, several challenges persist. Identifying cancer-type-specific methylation markers and addressing confounding effects from non-cancer-related methylation changes, such as those associated with aging and smoking, remain critical tasks, as these factors may diminish differences between patients with cancer and controls [29]. Therefore, limitations in both sensitivity and specificity may arise. In fact, urinary ctDNA methylation levels did not differ significantly in the current study, highlighting the challenges of using urine-based liquid biopsy for PDAC detection. These findings contribute to the growing body of evidence supporting methylation-based liquid biopsy as a non-invasive diagnostic tool [9] while underscoring the need for further research to optimize methodologies for urine-based detection.

Previous studies have suggested that renal function may influence ctDNA concentrations and biomarker detectability in urine, raising concerns about its impact on diagnostic accuracy. However, Terasawa et al. [16] reported that differences in renal function did not significantly affect ctDNA detectability in urine, suggesting that the lack of significant urine-based methylation differences in this study may be attributed to low tumor-derived ctDNA levels rather than renal clearance mechanisms.

In this study, eGFR levels were significantly lower in the non-cancerous group compared to the patients with PDAC, which could have influenced ctDNA clearance and biomarker levels in urine. Despite this difference, the absence of significant methylation differences in urine ctDNA suggests that factors beyond renal function, such as the biological characteristics of PDAC and the limited release of tumor-derived DNA into urine, may play a more crucial role.

### 3.2. Significance of Whole-Genome Bisulfite Sequencing (WGBS) in This Study

Although WGBS was utilized as a high-resolution approach for genome-wide methylation profiling, balancing the advantages of this method with its limitations remains essential when considering clinical applications. WGBS offers a comprehensive view of single-base methylation changes, which is crucial for identifying novel biomarkers. However, high costs, computational complexity, and the requirement for a large amount of input DNA pose challenges to routine clinical implementation [20,30]. In particular, the time and cost required for WGBS remain major barriers to its adoption in diagnostic workflows. While WGBS remains valuable for comprehensive discovery, future efforts should focus on translating identified differentially methylated regions (DMRs) into focused, cost-effective targeted methylation panels. Such panels could significantly improve throughput and affordability, making them more feasible for routine clinical use. Interpretation of the study results should take these considerations into account. Future research should explore the feasibility of using targeted methylation panels or adopting machine learning-driven approaches to enhance clinical applicability while maintaining diagnostic accuracy [31,32].

### 3.3. Biological Significance of Intergenic Methylation in PDAC

A key finding of this study showed that the majority of the significantly differentially methylated CpG sites were located within intergenic regions. This observation aligns with previous studies suggesting that intergenic DNA methylation is actively regulated in cancer, extending beyond simple gene silencing. Long non-coding RNAs (lncRNAs) and enhancer regions embedded within these intergenic domains may influence chromatin architecture, transcription factor binding, and gene expression in PDAC cells [33,34]. Additionally, global hypomethylation in intergenic regions has been associated with genomic instability and activation of oncogenic pathways [35,36]. Future studies should investigate whether these differentially methylated regions (DMRs) contribute to functional alterations in pancreatic tumorigenesis and whether they can be leveraged as diagnostic or therapeutic targets.

### 3.4. Challenges and Potential Strategies for Urine-Based Liquid Biopsy

Although urine-based liquid biopsy offers clear advantages in terms of noninvasiveness and ease of repeat sampling, this study found no significant differences in ctDNA methylation profiles between the patients with PDAC and the controls. Several factors may explain these findings:Low ctDNA concentration in urine: Insufficient levels of tumor-derived ctDNA in urine may hinder reliable detection, particularly for non-urological malignancies such as PDAC [10,37].Different ctDNA shedding mechanisms: Although plasma ctDNA is largely derived from apoptotic and necrotic tumor cells, urine ctDNA reflects transrenal passage, which may affect methylation patterns [38].Technical limitations: current methodologies for ctDNA extraction and bisulfite conversion may lack optimization for urine-based assays [11].

This study evaluated several extraction kits and selected one that reproducibly yielded sufficient ctDNA suitable for WGBS. However, alternative enrichment or concentration techniques not examined in this analysis may offer further improvements in ctDNA recovery from urine. Future studies should explore and validate various ctDNA processing protocols to enhance sensitivity.

To overcome these challenges, future studies should consider integrating multiple biomarkers, such as combining DNA methylation with mutation analysis or exosomal RNA profiling, to enhance diagnostic sensitivity [39]. Exploring machine learning-based classification algorithms may also improve differentiation between cancer and non-cancer samples [12]. Furthermore, standardizing the pre-analytical processing methods is crucial for optimizing ctDNA recovery from urine samples and ensuring reproducibility across studies [11].

### 3.5. Future Directions and Clinical Implications

The encouraging results regarding plasma ctDNA methylation warrant further validation in larger, multi-center studies to establish consistency across diverse pancreatic cancer populations. Emphasizing targeted methylation panels, longitudinal monitoring of methylation changes, and integration with other liquid biopsy techniques like mutation-based assays, exosomal RNA, and protein markers may enhance both diagnostic sensitivity and specificity. The exclusive focus on methylation in this study represents a limitation. Future research should prioritize multi-omic approaches to reduce false-negative rates and improve classification accuracy, ultimately enhancing the clinical utility of ctDNA methylation for early detection and disease monitoring in pancreatic cancer.

### 3.6. Limitations

This study had several limitations. First, the sample size was relatively small (*n* = 35 patients with PDAC and *n* = 10 non-cancer controls), which may limit the generalizability of the findings. Therefore, larger, well-defined cohorts are required for validation. Additionally, the study design (retrospective vs. prospective) was unclear, which affected data interpretation. Another limitation is the potential selection bias. The patient recruitment criteria were not explicitly controlled, indicating that tumor stage distribution, preexisting conditions, and demographic factors may have influenced the results. The retrospective nature of the study, which relied on available biospecimens, prevented predefined matching of cohorts based on sample size or sex ratio. This imbalance is acknowledged, and future research should implement stratified recruitment strategies to reduce demographic bias. Although plasma ctDNA showed significant methylation differences, urine ctDNA did not, suggesting inherent limitations in urine-based PDAC detection. Contributing factors include ctDNA degradation, low tumor-derived DNA content, and differences in ctDNA methylation dynamics between plasma and urine. Future research should explore ultra-sensitive sequencing techniques and multi-marker panels to improve urine-based detection. The non-cancer control group consisted of patients with benign or inflammatory pancreaticobiliary diseases, including cholelithiasis and cholecystitis, rather than healthy volunteers. This approach aimed to simulate a real-world differential diagnostic setting; however, comparison with studies that used healthy controls may be limited. Moreover, the exclusion of patients with chronic or tumor-forming pancreatitis—a critical differential in PDAC diagnosis—resulted from the unavailability of samples during the study period. Future inclusion of these patient groups will be essential for enhancing the discriminatory accuracy of methylation-based diagnostics. Although this study identified DMRs, their biological significance remained unclear. Further functional validation is required to determine whether these epigenetic changes play a role in tumorigenesis or serve as diagnostic markers. Moreover, correlation analysis between tumor tissue and ctDNA methylation profiles could not be conducted in this study, as tumor specimens were available for only 13 cases, which was insufficient for meaningful statistical comparison. Future studies should incorporate larger numbers of matched tissue and plasma samples to better elucidate the relationship between tumor-derived and circulating methylation signatures.

## 4. Materials and Methods

### 4.1. Patient Enrollment

All patients were diagnosed at Juntendo University Shizuoka Hospital between 2019 and 2022. Table 1 presents the clinical and demographic characteristics of the study cohort, which comprised 35 patients with PDAC and 10 non-cancerous controls. The mean age in both groups was 72.3 years, with no significant differences in age distribution (*p* = N.S.) or sex composition (*p* = N.S.). The estimated glomerular filtration rate (eGFR) was significantly lower in the non-cancerous group (59.7 ± 12.7 mL/min/1.73 m^2^) compared to the patients with PDAC (74.4 ± 20.4 mL/min/1.73 m^2^; *p* = 0.01).

Among the patients with PDAC, serum tumor markers were evaluated, yielding mean values of carcinoembryonic antigen (CEA) at 13.3 ± 24.5 ng/mL, CA19-9 at 1559.6 ± 4371.9 U/mL, DUPAN-2 at 1150.0 ± 2890.9 U/mL, and Span-1 at 870.5 ± 1863.2 U/mL. Tumor marker data were not available for the non-cancerous group. Various diagnostic methods were used among the patients with PDAC: EUS-FNA cytology in two cases, EUS-FNA biopsy in three cases, pancreatic juice cytology in one case, ascitic fluid cytology in one case, and surgical specimens in ten cases. Imaging techniques such as computed tomography (CT) and/or positron emission tomography (PET) were used for the remaining 18 cases. The cohort included patients across a broad range of disease stages: eight classified as clinical stage I, four as stage II, nine as stage III, and fourteen as stage IV.

The non-cancerous group included patients diagnosed with cholelithiasis (*n* = 5), cholecystitis (*n* = 4), and choledocholithiasis (*n* = 1), with none presenting malignant disease. Importantly, this control group consisted of individuals with benign or inflammatory pancreaticobiliary conditions rather than healthy volunteers, reflecting a clinically realistic differential diagnostic setting.

The study protocol adhered to the ethical guidelines of the World Medical Association and the Declaration of Helsinki. This study was approved by the Ethics Committee of Juntendo University School of Medicine (approval number: S19-0685). All patients provided informed consent for the use of their samples in scientific research.

### 4.2. Sample Processing

Plasma isolation began within 4 h after blood collection to preserve ctDNA integrity. Blood samples were centrifuged at 3000 rpm for 15 min at 4 °C, and the resulting supernatant was collected. Urine samples were similarly centrifuged, and the supernatant was collected.

### 4.3. Extraction, Bisulfite Conversion Library Preparation, and Sequencing

For WGBS, ctDNA was extracted from 3 mL of plasma using the Quick-ctDNA Serum & Plasma Kit (#D4076, ZYMO Research, Irvine, CA, USA) following the manufacturer’s protocol. From urine, ctDNA was extracted from 20 mL using the Quick-DNA Urine Kit (#D3061, ZYMO Research) as recommended by the manufacturer. Bisulfite conversion and library preparation of the extracted ctDNA were conducted using the Zymo-Seq™ Cell-Free DNA WGBS Library Kit (#D5462, ZYMO Research). Index PCR amplification was performed for 10 cycles. Adapter ligation and purification were performed using the Select-a-Size DNA Clean & Concentrator Magbead Kit (#D4085, ZYMO Research). Following PCR, samples were assessed using a bioanalyzer (Agilent 2100, Agilent Technologies, Santa Clara, CA, USA), and DNA concentrations were adjusted to 1 nM. Sequencing was performed according to the manufacturer’s protocol using the NextSeq 500 High-Output (150 cycles) V2.5 (#20024907, Illumina, San Diego, CA, USA). The denatured library mix was prepared at a final concentration of 1.0 pM and sequenced using the sequencer (NextSeq 500, Illumina).

### 4.4. Analysis of Sequence Data

After base recognition, all paired-end FASTQ files were processed using Cutadapt (v 1.8.3) to remove adapter sequences and low-quality bases (base quality below Q20; minimum read length: 36 bp). The hg38 (p16) human reference genome was obtained from the UCSC database (https://hgdownload.soe.ucsc.edu/downloads.html, accessed on 23 August 2023). Reads were aligned to the reference genome using Bowtie2 (v2.5.1) with default parameters. The resulting BAM files, generated by Bismark (v0.22.3), were sorted using Samtools (v1.18). The genome-wide cytosine methylation profiles were analyzed using the R packages methylKit (v1.26.0) and Genomation (v1.32.0) [40].

### 4.5. Statistical Analysis

Data analysis was performed using JMP 18.1.0 (SAS Institute Inc., Cary, NC, USA). Results are expressed as means ± standard deviation (SD). Measurement data were analyzed using Student’s *t*-test, and categorical data were assessed using the Chi-square test. Statistical significance was set at *p* < 0.05.

## 5. Conclusions

This study highlights the potential of plasma ctDNA methylation as a non-invasive biomarker for PDAC detection, with significant differences observed between patients with and without cancer. However, urine ctDNA methylation profiles did not show statistical significance, indicating that plasma is a more reliable biofluid for methylation-based diagnostics in PDAC. These findings have significant clinical implications. The identification of methylation markers in plasma ctDNA could lead to novel liquid biopsy-based diagnostic tools, providing a less invasive alternative to tissue biopsy. Although urine-based liquid biopsy remains an attractive approach owing to its ease of collection, further technological advancements and larger studies are required to improve its diagnostic accuracy.

Future studies should focus on validating these biomarkers in larger, multi-center cohorts, integrating machine learning for improved classification and evaluating their prognostic value in treatment monitoring. Ultimately, integrating methylation-based liquid biopsy techniques into clinical practice could enhance early PDAC detection, improving patient outcomes through timely and targeted interventions.

## Figures and Tables

**Figure 1 ijms-26-04972-f001:**
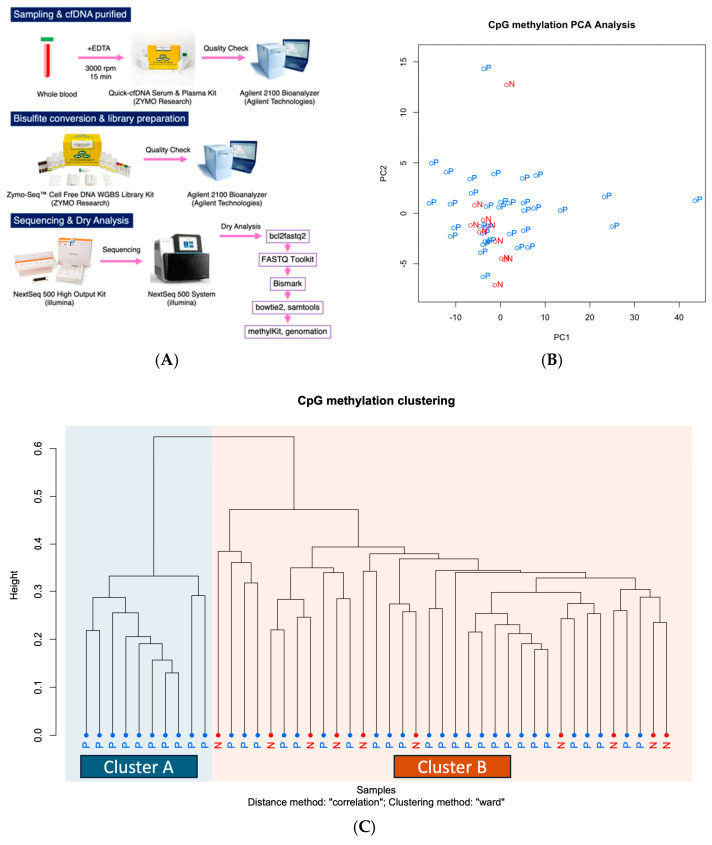
Methylation profile of ctDNA in plasma. (**A**) Workflow from sample collection to data analysis. (**B**) Principal component analysis (PCA) of plasma ctDNA methylation profiles. The first two components—PC1 and PC2—explained 24.5% and 3.4% of the total variance, respectively. Each point represents an individual sample, with colors indicating disease status: red for normal (N) and blue for pancreatic ductal adenocarcinoma (P). The separation observed between groups suggests that cancer and control samples exhibit distinct methylation profiles. (**C**) Clustering analysis of methylation profiles of ctDNA in plasma, comparing patients with (P) and without cancer (N).

**Figure 2 ijms-26-04972-f002:**
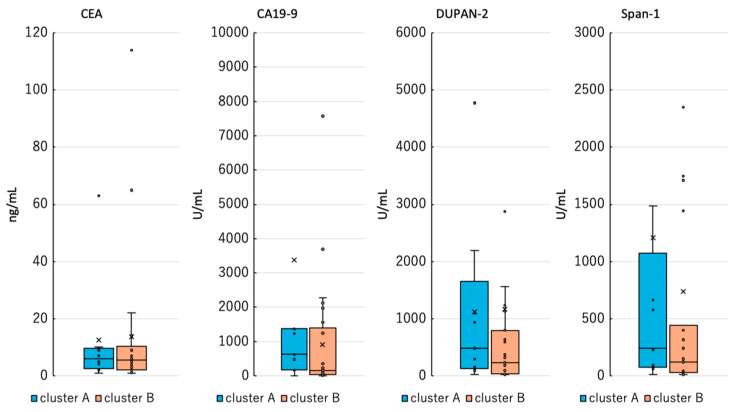
Comparison of existing tumor markers between cluster A and cluster B. Cluster B included tumor markers from patients with cancer only, excluding non-cancer controls.

**Figure 3 ijms-26-04972-f003:**
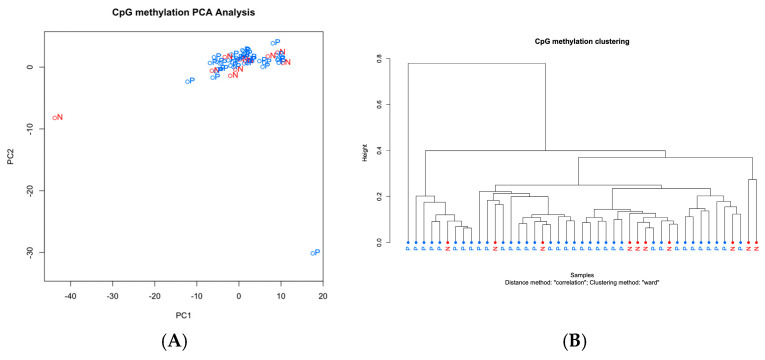
Methylation profile of ctDNA in urine. (**A**) Principal component analysis (PCA) of urine ctDNA methylation profiles. The first two components—PC1 and PC2—explained 30.5% and 9.4% of the total variance, respectively. Each point represents an individual sample, with colors indicating disease status: red for normal (N) and blue for pancreatic ductal adenocarcinoma (P). The separation observed between the groups suggests that cancer and control samples exhibit distinct methylation profiles. (**B**) Clustering analysis of methylation profiles of ctDNA in plasma, comparing patients with (P) and without cancer (N).

**Table 1 ijms-26-04972-t001:** Patient characteristics.

Characteristics	Pancreatic Ductal Adenocarcinoma Patients (*n* = 35)	Non-Cancerous Patients (*n* = 10)	*p*-Value
Age (years old), mean ± SD	72.3 ± 10.5	72.3 ± 5.7	N.S.
Sex (male/female)	18/17	8/2	N.S.
eGFR (mL/min/1.73 m^2^), mean ± SD	74.4 ± 20.4	59.7 ± 12.7	0.01
CEA (ng/mL), mean ± SD	13.3 ± 24.5	N/A	
CA19-9 (U/mL), mean ± SD	1559.6 ± 4371.9	N/A	
DUPAN-2 (U/mL), mean ± SD	1150.0 ± 2890.9	N/A	
SPan-1 (U/mL), mean ± SD	870.5 ± 1863.2	N/A	
**Method of final diagnosis**			
EUS-FNA cytology	2		
EUS-FNA biopsy tissue	3		
Cytology of pancreatic juice	1		
Cytology of ascites	1		
Surgical specimen	10		
CT and/or PET	18		
**Clinical stage (cStage)**			
cStage I	8		
cStage II	4		
cStage III	9		
cStage IV	14		
**Diagnosis for non-cancerous patients**			
Cholelithiasis		5	
Cholecystitis		4	
Choledocholithiasis		1	

SD, standard deviation; eGFR, estimated glomerular filtration rate; CEA, carcinoembryonic antigen; CA19-9, carbohydrate antigen 19-9; EUS-FNA, endoscopic ultrasound-guided fine-needle aspiration; N/A, not available; N.S., not significant.

## Data Availability

Data is contained within the article and Appendix A.

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
