# Peer review of "Plasma and Urine Circulating Tumor DNA Methylation Profiles for Non-Invasive Pancreatic Ductal Adenocarcinoma Detection: Significant Findings in Plasma Only"

_ijms, 2025, doi:10.3390/ijms26114972_

Round 1
Reviewer 1 Report
Comments and Suggestions for Authors
The aim of this study was to explore methylation profiles of circulating tumor DNA (ctDNA) in plasma and urine as biomarkers for noninvasive detection of pancreatic ductal adenocarcinoma (PDAC). The results suggest that plasma ctDNA methylation has potential as a diagnostic marker for PDAC, while urine ctDNA may not be suitable for liquid biopsy of PDAC due to limitations in detection sensitivity and biological mechanisms. However, the article has the following problems:
- It is suggested to supplement the explanatory note of PCA analysis and optimize the typesetting.
- There is duplication of secondary heading numbers in the discussion section;
- It is suggested to add the literature on detection of other non-urinary cancers based on urine ctDNA methylation profile in recent three years to support the conclusion that "urine ctDNA is not significantly different in non-urinary cancers";
- Combination analysis of methylation markers with other PDAC fluid biopsy markers can be added to assess the efficacy of the combination assay;
- Different ctDNA enrichment methods can be tried to optimize the processing of urine samples.
- Why are there large differences in sample size and sex ratio between PDAC patients and non-cancer controls?
- Are there differences in the ctDNA methylation profile of non-cancer patients with biliary diseases and healthy individuals?
- Are there differences in plasma ctDNA methylation profiles between PDAC patients and pancreatitis patients?
I haven't found any major issues with the English language. Please carefully check the format, etc.
Author Response
Dear Reviewer
We sincerely thank you for your thorough evaluation of our manuscript entitled "Plasma and Urine Circulating Tumor DNA Methylation Profiles for Non-Invasive Pancreatic Ductal Adenocarcinoma Detection: Significant Findings in Plasma Only." We greatly appreciate your constructive and insightful comments, which have significantly improved the quality and clarity of our work.
We have revised the manuscript accordingly, with all changes marked in the red font. Below, we provide a detailed, point-by-point response to each comment. We hope the revised manuscript now meets the standards required for publication.
Comment 1:
“It is suggested to supplement the explanatory note of PCA analysis and optimize the typesetting.”
Response:
We appreciate the reviewer’s helpful suggestion. In the revised manuscript, we have supplemented the explanatory note for the PCA analysis (see page 5, lines 160–163 and page 6, lines 201–205). Specifically, we clarified the data used for PCA, the proportion of variance explained by each component, and the interpretation of clustering patterns. Additionally, we optimized the figure typesetting by adjusting font sizes, axis labels, and figure resolution to enhance overall readability. Furthermore, we replaced Figure 1B and Figure 3A with higher-quality versions to improve visual quality and interpretability.
Comment 2:
“There is duplication of secondary heading numbers in the discussion section.”
Response:
We apologize for this oversight. The duplicated secondary heading numbers in the discussion section have been corrected to ensure consistent and logical formatting throughout the manuscript (see page 8, line 297).
Comment 3:
“It is suggested to add the literature on detection of other non-urinary cancers based on urine ctDNA methylation profile in recent three years to support the conclusion that 'urine ctDNA is not significantly different in non-urinary cancers'.”
Response:
We appreciate this valuable suggestion. In response, we have cited recent studies that explore the use of urine ctDNA methylation profiling in non-urological cancers, such as cervical, colorectal, endometrial, liver, lung, and ovarian cancers (see page 6, lines 211-215 and 224–230; references [22–29] in the revised reference list). Although these studies demonstrate the feasibility of urine-based methylation analysis, most report potential utility rather than consistent statistical significance across large cohorts. Therefore, we have emphasized in the revised Discussion section that urine ctDNA methylation analysis holds promise, particularly due to the biological characteristics of DNA methylation. Simultaneously, we have elaborated on current limitations, including low tumor-derived DNA concentrations in urine, interference from background DNA, and confounding effects due to non-cancerous conditions such as aging or inflammation. These elements collectively justify further refinement before reliable application to non-urological cancers can be achieved.
Comment 4:
“Combination analysis of methylation markers with other PDAC fluid biopsy markers can be added to assess the efficacy of the combination assay.”
Response:
We agree with this insightful suggestion. In the revised manuscript, we have added a discussion on the potential advantages of combining ctDNA methylation profiles with other biomarkers, such as mutation-based assays, exosomal RNA, or protein markers for PDAC detection (see page 8, lines 290–295).
Comment 5:
“Different ctDNA enrichment methods can be tried to optimize the processing of urine samples.”
Response:
Thank you for this valuable suggestion. In the revised discussion, we now acknowledge that although a reproducible extraction protocol was used, future studies should explore alternative ctDNA enrichment and concentration techniques to enhance the sensitivity of urine-based ctDNA analysis (see page 7, lines 278–281).
Comment 6:
“Why are there large differences in sample size and sex ratio between PDAC patients and non-cancer controls?”
Response:
We appreciate the reviewer’s observation. The noted differences in sample size and sex ratio stem from the retrospective design of the study and reliance on archived samples. As noted in the revised limitations section (page 8, lines 303–305), future research should implement stratified recruitment strategies to reduce demographic bias.
Comment 7:
“Are there differences in the ctDNA methylation profile of non-cancer patients with biliary diseases and healthy individuals?”
Response:
We thank the reviewer for this insightful question. The non-cancer control group consisted of patients with benign or inflammatory pancreaticobiliary diseases, including cholelithiasis and cholecystitis, rather than healthy volunteers. This approach aimed to simulate a real-world differential diagnostic setting. We have clarified this point in the Methods and Discussion sections (see page 3, lines 92–94 and page 8, lines 309–312 of the revised manuscript).
Comment 8:
“Are there differences in plasma ctDNA methylation profiles between PDAC patients and pancreatitis patients?”
Response:
We appreciate the reviewer’s important observation. Although our study targeted a real-world diagnostic setting, we acknowledge that including patients with chronic or tumor-forming pancreatitis would have provided a more comprehensive comparison. Due to limitations in sample availability during the collection period, such cases were not included. This limitation has now been explicitly stated in the revised manuscript (see page 8, lines 312–314), and we propose this comparison as an essential direction for future research.
Reviewer 2 Report
Comments and Suggestions for Authors
The author analyzed the methylation state of circulating free DNA in the blood and urine of pancreatic ductal adenocarcinoma patients and compared it with that of controls. Based on the difference between the blood samples from the patients and controls, they propose methylation analysis as a diagnostic tool.
The work is preliminary because of the small sample size. However, it is interesting enough to be published.
There are several issues to be addressed before publication.
One issue is that the analysis detects only a part of the patients. Only 10 patients among 35 are classified in a positive group, while others are not distinguishable from the controls. Although there is no false positive case, many false negative cases exist. The authors should mention this point in the text.
They should also analyze and discuss if there is some tendency for the patients to be classified in the positive group (sex, stage, etc.), even if the number of cases is too small to draw a conclusion.
It will be interesting to analyze the methylation status of DNA in cancer cells of the patients and examine if there is a correlation with that of blood samples.
Another point is that the number of female controls is too small. They should note this point, too.
They should also discuss the cost and time for the analysis. This is important for the development of the analysis as a diagnostic test.
Author Response
Dear Reviewer,
We sincerely thank you for your thorough evaluation of our manuscript entitled "Plasma and Urine Circulating Tumor DNA Methylation Profiles for Non-Invasive Pancreatic Ductal Adenocarcinoma Detection: Significant Findings in Plasma Only." We greatly appreciate your constructive and insightful comments, which have significantly improved the quality and clarity of our work.
We have revised the manuscript accordingly, with all changes marked in the red font. Below, we provide a detailed, point-by-point response to each comment. We hope the revised manuscript now meets the standards required for publication.
Comment 1:
“Only 10 patients among 35 are classified in a positive group, while others are not distinguishable from the controls. Authors should mention this point.”
Response:
We appreciate this important observation. Indeed, only a subset of patients with PDAC showed detectable methylation differences. We now emphasize in the Discussion that “Although high specificity was observed in plasma ctDNA methylation, sensitivity remained limited, as only a subset of patients with PDAC exhibited detectable changes. This finding may reflect biological heterogeneity or low ctDNA shedding in some cases, suggesting the occurrence of false-negative results. Indeed, only 10 out of 35 patients with PDAC were classified as positive based on methylation profiling, indicating that the current approach identifies only a subset of true cases. Despite the absence of false positives, the high number of false negatives highlights the sensitivity limitations. These findings underscore the need for incorporating additional biomarkers or adopting multi-omic strategies to improve detection rates.” This limitation is discussed in the revised manuscript (page 6, lines 217–223).
Comment 2:
“They should also analyze and discuss if there is some tendency for the patients to be classified in the positive group (sex, stage, etc.), even if the number of cases is too small to draw a conclusion.”
Response:
Thank you for this thoughtful suggestion. We performed an exploratory analysis, which revealed no apparent trends in sex, stage, age, or tumor size between clusters. We have added this information to the revised manuscript (page 5, lines 171–172).
Comment 3:
“It will be interesting to analyze the methylation status of DNA in cancer cells of the patients and examine if there is a correlation with that of blood samples.”
Response:
We agree with the reviewer that this would provide valuable insights. However, due to the limited availability of corresponding tumor tissue samples (n = 13), we could not perform meaningful comparisons. This limitation has been acknowledged in the revised manuscript (page 8, lines 317–321).
Comment 4:
“Another point is that the number of female controls is too small. They should note this point, too.”
Response:
We thank the reviewer for highlighting this point. The sex imbalance—particularly the small number of female controls—has now been explicitly noted as a limitation of the study. This has been added to the revised manuscript (page 8, lines 303–305).
Comment 5:
“They should also discuss the cost and time for the analysis. This is important for the development of the analysis as a diagnostic test.”
Response:
We appreciate this important suggestion. A discussion regarding the high cost and processing time associated with WGBS and the potential for translating our findings into targeted panels for clinical use has been added in the revised manuscript (page 7, lines 250–253).
Round 2
Reviewer 1 Report
Comments and Suggestions for Authors
The author answered all my questions and made revisions in the revised manuscript. Several of the defects acknowledged by the authors are elaborated in the limitations section. I am satisfied with the revised manuscript.